# Nitrogen, phosphorus and potassium formula fertilization model of *Lonicera japonica Thunb* in hilly and gully Loess Plateau of China

Xiaofeng Jiang[1,2,3,4]*, Jianguo Guo[1]*, Shangzhong Li[1,2,3,4], Yuan Chen[5], Zhaohui Shi[6], Ping Xu[6], Bo Dong[1,2,3,4], Jiali Chen[5], Qing Fang[5], Yusheng Zhai[6]

**1** Dryland Agriculture Institute/ Institute of Plant Protection, Gansu Academy of Agricultural Sciences, Lanzhou, Gansu, China, **2** Key Laboratory of High Water Utilization on Drvland of Gansu Province, Lanzhou, Gansu, China, **3** Key Laboratory of Low-carbon Green Agriculture in Northwestern China, Ministry of Agriculture and Rural Affairs, Lanzhou, Gansu, China, **4** The Jiont Key Laboratory of Agriculture and Rural Affairs-Gansu Province for Crop Drought Resistance, Yield Increment and Rainwater Efficient Utilization on Rain-fed Area, Lanzhou, Gansu, China, **5** Agronomy College, Gansu Agricultural University, Lanzhou, Gansu, China, **6** Tongwei Qingliang Yuan Limited Liability Company of Lonicerae Japonicae Flos Industry Poverty Alleviation and Development, Dingxi, Gansu, China

* jguo1001@163.com (JG); jxf_5188@163.com (XJ)

## Abstract

Formula fertilization has been shown to effectively increase both yield and economic profit in medicinal plants. *Lonicera japonica Thunb (L. japonica)* is a significant medicinal plant; however, no studies have yet investigated the impact of formula fertilization on its performance. Therefore, the objective of this study is to establish models for enhancing the yield of *L. japonica* through formula fertilization using various ratios of nitrogen (N), phosphorus (P), and potassium (K) in the primary production areas of the hilly and gully Loess Plateau in China. A '3414' formula fertilization trial was conducted, examining different doses and ratios of N, P, and K applied to *L. japonica*.Fertilizer was applied to a small pit, 30 cm deep, surrounding the root of each tree. The spacing between plants and rows was maintained at 1.0 m x 1.0 m. A fertilizer model was developed by calculating and comparing the yield and economic profit of the plants. The results indicated that the application of nitrogen (N), phosphorus (P), and potassium (K) in the soil significantly improved yield. Phosphorus (P) was identified as the primary limiting factor affecting yield. The output-to-investment ratio of the $N_2P_2K_3$ fertilizer was significantly reduced. The fertilizer effect function can be expressed as: $Y = 296.66 - 0.81X_1 + 11.53X_2 + 4.05X_3 - 0.051X_1^2 - 0.60X_2^2 - 0.28X_3^2 + 0.13X_1 \times X_2 + 0.20X_1 \times X_3 - 0.029X_2 \times X_3$. The recommended doses of nitrogen (N), phosphorus (P), and potassium (K) fertilizers are 18.25–26.75 g, 6.27–8.73 g, and 10.04–13.96 g per plant, respectively.The application of formula fertilizer in this study resulted in a notable increase in yield and economic benefit, with improvements of 38.18% and an additional 0.3729 USD per plant, respectively. These findings significantly contribute to the production of *L. japonica,* enhancing both yield and profitability in specific geographical conditions suited for cultivating economic plants.

**Data availability statement:** All relevant data are publicly available from the Figshare repository at the following DOI: https://doi.org/10.6084/m9.figshare.28826912.

**Funding:** This work was supported by multiple funding sources. The National Key Research and Development Project and its sub-projects (Grant Numbers: 2021YFD1100504, 2021YFD1100504-1, 2021YFD1100504-2) provided significant support in terms of personnel allocation, technical assistance, and financial resources. The Agricultural Science and Technology Independent Innovation Special Key Research and Development Program of Gansu Academy of Agricultural Sciences (Grant Number: 2021GARS11) contributed through the sharing of technical exchanges and experimental infrastructure. Additional support was received from the Gansu Province Science and Technology Plan Project: Livelihood Science and Technology Special – Topics on Rural Revitalization (Grant Number: 22CX2NA007), and the National Natural Science Foundation of China (NSFC) (Grant Number: 31460547). This research was also funded by the Special Fund for Scientific Innovation Strategy – Construction of High-level Academy of Agriculture Science (Grant Number: GNKJ-2021-33). All grants were awarded to Professor Xiaofeng Jiang. The funders had no role in study design, data collection and analysis, decision to publish, or preparation of the manuscript.

**Competing interests:** The authors have declared that no competing interests exist.

## Introduction

*Lonicera japonica Thunb* (*L. japonica*), also known as honeysuckle, belongs to the *Caprifoliaceae* family [1]. Gansu Province is the main producing area of most authentic Chinese medicinal herbs in China. Their buds are harvested and dried before bloom in early summer, which are rich in organic acids, flavonoids, terpenoids, and volatile oils, so they have been widely recognized as effective medicinal herbs for heat-clearing and detoxification, and are commonly used in clinical practice for treating various diseases including pneumonia [2]. Recently, the market demand for honeysuckle is increasing significantly due to the impact of COVID-19 [3,4]. As a result, cultivation has become the primary method of obtaining raw materials [5,6]. However, research on efficient cultivation techniques for honeysuckle is still in its early stages. Therefore, high-quality and efficient production of *L. japonica* is in urgent need to overcome traditional improper fertilization practices.

Formula fertilization may be an effective way to improve yield and quality of *L. japonica* [7,8]. Former research found that combined application on N, P, and K could not only promote the biosynthesis of chlorogenic acid and flavonoids in *L. japonica*, but also increase the yield and quality [9–12]. For example, the application of nitrogen fertilizer in Japan enhanced the vegetative growth of *L. japonica*, increased the yield of vegetative organs, and elevated the crude protein content of the leaves. However, it resulted in a decrease in both economic yield and chlorogenic acid content of the buds of the reproductive organs [7,13–15]. Adding P could not only promote flower bud differentiation and increase economic yield, but also increase the Chlorogenic acid content of flower buds and improve commodity quality [16–18]. When the recommended dosages of N, $P_2O_5$, and $K_2O$ were 34.9–47.9, 13.1–14.4, 18.5–23.8 g/plant, the economic yield of *L. japonica* was 387.0–430.3 g/plant [19].

Tongwei County locates in the hilly and gully region of the Loess Plateau in central Gansu Province. The landform in the territory is composed of hilly and gully like hills, residual tableland. The mountain ridges are undulating, and the ravines are criss-crossing [20]. The annual sunshine duration are 1400 h, and the annual total solar radiation is 5446.5 MJ/m². There are abundant light and heat resources, which are conducive to the development of modern Silk Road agricultural practices in cold and arid areas. Currently, a high-end *L. japonica* planting base spanning 6,667 ha has been established within the Tongwei Qingliang Yuan Limited Liability Company. This endeavor is part of the Poverty Alleviation and Development initiative in the *L. japonica* Industry [21–23]. Moreover, the produced *L. japonica* has achieved a remarkable status, both in terms of quality and industry reputation, being at the forefront within China. However, due to long-term planting, there has been a significant increase in fertilizer usage for per tree, which has become a major hindrance to the development of local industries. There is an urgent need to investigate the optimal dosage of N, P, and K fertilizers for *L. japonica*. Therefore, this study conducted an experiment using '3414' Fertilizer Design experiment to establish a model and finally determine a optimal formula for cultivating *L. japonica* with high yield and economic profit.

## Materials and methods

### Overview of the test site

The experiment began in April 2022. The experimental site located in Wangjiahe Community, Yaochuan Village, Lidian Township, Tongwei County (1860 m above sea level, 35 ° 25 ′ 40 ″ N, 105 ° 10 ′ 30 ″ E). It belongs to temperate semi-arid Continental climate type, with four distinct seasons, drought and little rain, and abundant light resources. The average annual temperature was 8.2°C, and the annual precipitation was 300 mm, most of them are from July to September. The frost free period was 145 d. The crops mature once a year and belong to the hilly and gully agricultural area of the Loess Plateau in central Gansu, The soil is loam. Irrigation was solely reliant on rainfall, with water, light, and other natural factors remaining consistent. During the growth period, artificial management techniques, including weeding and uniform pesticide spraying, were employed. Imidacloprid 25% wettable powder were sprayed at rate of 24 g a.i. per hectare for the plant.

The experimental site for this experiment was an authorized experimental base of Gansu Agricultural Academy, which has been approved and authorized by the provincial government to serve as an experimental field site. The development of *L. japonica* industry has unique natural advantages. The soil pH of the experimental site was 8.49. The organic matter in soil was 11.01 g/kg. total N was 0.6 g/kg, available N was 42.80 mg/kg. Total P was 0.72 g/kg, available P was 8.97 mg/kg, total K was 17.15 g/kg, the available K was 160.67 mg/kg, The cation exchange capacity of the soil was measured at 8.69 cmol/kg. The granulometric composition of the soil was as follows: 4.0% for particles sized between 0.2 and 2 mm, 48.1% for particles sized between 0.02 and 0.2 mm, 34.3% for particles sized between 0.002 and 0.02 mm, and 13.6% for particles smaller than 0.002 mm.

### Materials

The tested *L. japonica* species was Beihua No.1 of 4 year's cultivating time. The fertilizers used in the test were mainly urea (total N content is 46%, Petro China Company Limited), granular Superphosphate (the effective $P_2O_5$ was 12%, Baoji Fenghe Chemical Co., Ltd.), Potassium sulfate ($K_2O$ was 50%, Qinghai Laishuo Industry and Trade Co., Ltd.).

### Method

**Design of experiments.** This experiment adopted quadratic regression optimal design scheme of '3414', The '3414' experiment is a testing methodology outlined in the technical specifications for soil testing and formula fertilization established by the Ministry of Agriculture and Rural Affairs of the People's Republic of China, aimed at studying the effects of fertilizers. The '3' in '3414' signifies three factors: N, P, and K, while '4' indicates four application levels for each nutrient. Specifically, the four levels for nitrogen fertilizer application are N0, N1, N2, and N3; for phosphorus, they are $P_0$, $P_1$, $P_2$, and $P_3$; and for potassium, they are $K_0$, $K_1$, $K_2$, and $K_3$.In this study, we categorize fertilization levels into four distinct categories: 'level 0' signifies no fertilization; 'level 2' represents the optimal local fertilization amount (N: 30g/plant, P: 10g/plant, K: 16g/plant); 'level 1' corresponds to 0.5 times the 'level 2' amount; and 'level 3' indicates 1.5 times the 'level 2' amount. Furthermore, the term '14' refers to the fourteen distinct fertilization treatments applied for nitrogen, phosphorus, and potassium, denoted as follows: $N_0P_0K_0$, $N_1P_2K_2$, $N_0P_2K_2$, $N_2P_0K_2$, $N_2P_1K_2$, $N_2P_2K_2$, $N_2P_3K_2$, $N_2P_2K_0$, $N_2P_2K_1$, $N_2P_2K_3$, $N_3P_2K_2$, $N_1P_1K_2$, $N_1P_2K_1$, and $N_2P_1K_1$ [12,24–26].The fertilizer was poured into a small pit, 30 cm deep, surrounding the roots of each tree. The fertilizer was then mixed evenly with the soil using a shovel before filling the pit and surrounding ground with soil.

This was the first time for formula fertilization experiment carried out on the Tongwei *L. japonica* in the hilly and gully area of the Loess Plateau in central Gansu. There is no reference data for the determination of the 2 levels of each factor (the approximate value of the local optimal fertilization amount). Therefore, it was determined by reference to the local yield level in 2016–2021 and the fertilization amount in Tongwei County. The fertilization dose was 30 g N, 10 g $P_2O_5$, and

16 g $K_2O$ per plant. Each treatment had 6 *L. japonica* trees, 3 replicates, arranged by random blocks, and the spacing between plants and rows was 1.0 mX1.0 m. Nitrogen fertilizer, phosphorus fertilizer, and potassium fertilizer were applied in a one-time manner on April 8, 2022 before germination. The method of fertilization was ditch application, and the fertilization amount was determined according to Table 1.

**Measurement items and methods.** Yield measurement: the total yield was weighted in June. The fresh and wet flower yield measurement was through the manual picking of workers, each person picked six trees, the flower weight of each tree was weighed with a high-precision electronic balance (model AE001, Jiangsu Hongli weighing Equipment Co., LTD.), and the yield of each tree was recorded. The average value of the flower yield of six trees was the yield of a single tree.

Soil total nitrogen content was determined by NY/T 53–1987 standard Kjeldt nitrogen determination method, soil total phosphorus content was determined by NY/T 88–1988 standard molybdenum-antimony resistance colorimetric method, and soil total potassium content was determined by NY/T 87–1988 standard acid melting flame photometer and FP6450 type flame photometer. The cation exchange capacity was determined using the EDTA ammonium acetate exchange method (pH > 7.5), as outlined in Chapter 3, Section 5, Method 2 of the Soil Analysis Technical Specification. Additionally, the granulometry of the soil was assessed following the HJ 1068–2019 guidelines for the determination of soil particle size, utilizing both the pipette method and the hydrometer method.

## Data analysis

DPS7.5 software and Exce1 2010 software were used to fit the univariate quadratic, binary quadratic and ternary quadratic fertilizer effect functions of yield and fertilizer application, and whether they comply with typical fertilizer efficiency models was analyzed. If it belongs to an atypical fertilizer efficiency model, the frequency analysis method was used to calculate the economic optimal fertilizer amount and the maximum yield fertilizer amount.

The frequency analysis calculation formula was [14,20]:

$$\overline{x}_i = \frac{1}{n_j} \sum_{j=0}^{p} u_j n_{ij} (i = 1, 2, 3)$$

(1)

**Table 1. Experimental design of '3414' test.**

| Number | Treatment | N-coded value | P-coded value | K-coded value | Fertilization amount (g/plant) | | |
|---|---|---|---|---|---|---|---|
| | | | | | N | $P_2O_5$ | $K_2O$ |
| 1 | $N_0P_0K_0$ | 0 | 0 | 0 | 0 | 0 | 0 |
| 2 | $N_0P_2K_2$ | 0 | 2 | 2 | 0 | 10 | 16 |
| 3 | $N_1P_2K_2$ | 1 | 2 | 2 | 15 | 10 | 16 |
| 4 | $N_2P_0K_2$ | 2 | 0 | 2 | 30 | 0 | 16 |
| 5 | $N_2P_1K_2$ | 2 | 1 | 2 | 30 | 5 | 16 |
| 6 | $N_2P_2K_2$ | 2 | 2 | 2 | 30 | 10 | 16 |
| 7 | $N_2P_3K_2$ | 2 | 3 | 2 | 30 | 15 | 16 |
| 8 | $N_2P_2K_0$ | 2 | 2 | 0 | 30 | 10 | 0 |
| 9 | $N_2P_2K_1$ | 2 | 2 | 1 | 30 | 10 | 8 |
| 10 | $N_2P_2K_3$ | 2 | 2 | 3 | 30 | 10 | 24 |
| 11 | $N_3P_2K_2$ | 3 | 2 | 2 | 45 | 10 | 16 |
| 12 | $N_1P_1K_2$ | 1 | 1 | 2 | 15 | 5 | 16 |
| 13 | $N_1P_2K_1$ | 1 | 2 | 1 | 15 | 10 | 8 |
| 14 | $N_2P_1K_1$ | 2 | 1 | 1 | 30 | 5 | 8 |

In the formula, $n_{ij}$ represents the number of occurrences of the variable value $X_{ij}$, where $(u_j)$, and $n_j$ represents the total number of occurrences of the variable $X_i$.

Standard deviation:

$$S\overline{x}_i = \frac{1}{\sqrt{n_j-1}} \cdot \sqrt{\sum_{j=0}^{p} (u_j - \overline{x}_i)^{\cdot 2} n_{ij}}$$

(2)

Mean standard deviation:

$$S_x = \frac{S}{\sqrt{n_j}}$$

(3)

95% confidence interval:

$$\left(\overline{x}_i - S\overline{x}_i \cdot t_{0.05(n_j-1)}\right) \sim \left(\overline{x}_i + S\overline{x}_i \cdot t_{0.05(n_j-1)}\right)$$

(4)

The economic benefits were calculated as follows:

Output value (CNY/plant) = Yield of per plant*24/1000

(5)

Fertilizer input (CNY/plant) = (nitrogen fertilizer amount of per plant *4.8/1000)
+ (phosphorus fertilizer amount of per plant *6/1000)
+ (potassium fertilizer amount of per plant *8/1000

(6)

Yield to investment ratio (%) = output value/fertilizer input

(7)

## Result

### N and P fertilizers promoting the increase of yield and economic benefits of *L. japonica* per plant

Soil fertility is the basic resource for sustainable agricultural development, nitrogen and phosphorus are the essential nutrients for crop growth and development, nitrogen and phosphorus affect the growth and development of crops, and then affect their yield and economic benefits [27]. Guo Yu et al.'s study on Qianhu found that reasonable distribution of soil nitrogen and phosphorus nutrients can effectively increase the accumulation of soil available nutrients and rhizosphere nutrients, and promote the increase of yield [27]. The index of soil nutrient abundance and deficiency was expressed as the percentage of yield in the nutrient deficient area to the total yield in fertilizer deficient area. That is, the relative yield can express the abundance and deficiency of soil nutrients [7]. Soil nutrients with a relative yield below 50% are extremely low; 50% -75% is low; 75% -95% is medium; greater than 95% is high. Thus, the index of soil nutrient abundance and deficiency of a certain crop in a certain area and the corresponding fertilizer amount were determined [19,20]. It is high to determine the soil nutrient abundance and deficiency index of a certain crop in a certain area and the corresponding fertilization amount. From Table 2, it can be seen that the contribution rate of basic soil fertility in the field trial site is 73.97%. The yield in the non fertilized area accounts for 73.97% of the total yield in the fully fertilized area, the yield in the N deficient area accounts for 84.67% of the total yield in the fully fertilized area, the yield in the P deficient area accounts for 78.70% of the total yield in the fully fertilized area, and the yield in the K deficient area accounts for 80.72% of the total yield in the fully fertilized area. The findings indicated that the soil's available nitrogen, phosphorus, and potassium content

fell within the moderate range, with the soil's capacity to supply these nutrients being at an intermediate level. Therefore, the P content of soil is the primary fertility factor that limits the yield of *L. japonica*. Within the recommended fertilization range,the application of 1 g each of nitrogen (N), phosphorus (P), and potassium (K) fertilizers can increase the yield of buds by 2.04 g, 8.52 g, and 4.82 g, respectively. The results indicate that increasing phosphorus fertilizer significantly enhances the yield of *Lonicera japonica* buds.

From Table 3, it can be seen that the yield of *L. japonica* per plant was greatly affected by different treatments. Among them, the yield and economic benefits of $N_2P_3K_2$ treatment were the highest at 409 g per plant and 1.3499 USD per plant. The treatment of $N_2P_3K_2$ increased the yield significantly by 38.18% compared to the control, while the yield and economic benefits of the non fertilized control were the lowest on 296 g per plant and 0.9770 USD per plant. The highest output to investment ratio was the treatment of $N_1P_2K_1$, which reached 47.26. However, the lowest output ratio for output to investment ratio was $N_2P_2K_3$, which was 23.96. For treatments of $N_0P_2K_2$, $N_1P_2K_2$, $N_2P_2K_2$, $N_3P_2K_2$, and $N_2P_0K_2$, $N_2P_1K_2$, $N_2P_2K_2$, and $N_2P_3K_2$ in Table 3,With the increase of nitrogen and phosphate fertilizer amount, the yield per plant of *L. japonica* increased gradually. From the treatments of $N_2P_2K_2$, $N_2P_2K_0$, $N_2P_2K_1$, and $N_2P_2K_3$ in From Table 2, it can be seen

**Table 2. The relative yield and increasing rate of fertilizer deficiency treatments.**

| Processing number | | Yield (g/plant) | Relative yield (%) | •Increased yield of 1g fertilizer production of bud fresh weight (g) | | |
|---|---|---|---|---|---|---|
| | | | | N | $P_2O_5$ | $K_2O$ |
| 1 | $N_0P_0K_0$ | 295.77 | 73.97 | – | – | – |
| 2 | $N_0P_2K_2$ | 338.53 | 84.67 | 2.04 | – | – |
| 4 | $N_2P_0K_2$ | 314.67 | 78.70 | – | 8.52 | – |
| 8 | $N_2P_2K_0$ | 322.73 | 80.72 | – | – | 4.82 |
| 6 | $N_2P_2K_2$ | 399.83 | 100.00 | – | – | – |

https://doi.org/10.6084/m9.figshare.28826912.

**Table 3. Effects of different treatments on *L. japonica* yield and economic benefits.**

| Experiment number | | Yield (g/plant) | Increased profit than control (%) | Economic profit (CNY/ plant) | Fertilizer cost (CNY/ plant) | Ratio of profit to investment |
|---|---|---|---|---|---|---|
| 1 | $N_0P_0K_0$ | 295.77 ± 2.85j | – | 7.10 | 0.000 | – |
| 2 | $N_0P_2K_2$ | 338.5 ± 2.21h | 14.46 | 8.12 | 0.188 | 43.22 |
| 3 | $N_1P_2K_2$ | 389.7 ± 5.30c | 31.76 | 9.35 | 0.260 | 35.97 |
| 4 | $N_2P_0K_2$ | 314.67 ± 9.08i | 6.39 | 7.55 | 0.272 | 27.76 |
| 5 | $N_2P_1K_2$ | 378.5 ± 10.41de | 27.97 | 9.08 | 0.302 | 30.08 |
| 6 | $N_2P_2K_2$ | 399.83 ± 8.13ab | 35.19 | 9.60 | 0.332 | 28.90 |
| 7 | $N_2P_3K_2$ | 408.70 ± 2.86a | 38.18 | 9.81 | 0.362 | 27.10 |
| 8 | $N_2P_2K_0$ | 322.73 ± 1.16i | 9.12 | 7.75 | 0.204 | 37.97 |
| 9 | $N_2P_2K_1$ | 372.27 ± 4.92ef | 25.86 | 8.93 | 0.268 | 33.34 |
| 10 | $N_2P_2K_3$ | 395.27 ± 1.68bc | 33.64 | 9.49 | 0.396 | 23.96 |
| 11 | $N_3P_2K_2$ | 407.93 ± 4.14a | 37.92 | 9.79 | 0.404 | 24.23 |
| 12 | $N_1P_1K_2$ | 367.80 ± 4.78f | 24.35 | 8.83 | 0.230 | 38.38 |
| 13 | $N_1P_2K_1$ | 386.73 ± 5.14 cd | 30.76 | 9.28 | 0.196 | 47.36 |
| 14 | $N_2P_1K_1$ | 353.4 ± 10.83g | 19.49 | 8.48 | 0.238 | 35.64 |

**Note:** The '-' indicates does not exist, the unit price of *L. japonica* is 3.3024 USD/kg, N is 0.6605 USD/ kg, $P_2O_5$ is 0.8256 USD/kg, and $K_2O$is 1.1008 USD/kg.

https://doi.org/10.6084/m9.figshare.28826912.

that as rate of P increases in production, the yield increased at low ratios and reached its highest value at treatment of No.6. However, when the amount of P application continued to increase until to treatment 10, the yield of per plant did not increase any more, but decreased slightly. The results of interaction relationship of fertilizer was PK>NP>NK, and the yield increased 14.46%, 9.12% and 6.39%, respectively compared with treatment with no fertilizer.

## Analysis of fertilizer effect function

**Analysis of ternary quadratic fertilizer effect function.** Using DPS software to perform regression analysis between independent variables x and y (yield per plant), the fertilizer effect function was calculated as: $Y = 296.66 - 0.81X_1 + 11.53X_2 + 4.05X_3 - 0.051X_1^2 - 0.60X_2^2 - 0.28X_3^2 + 0.13X_1{}^*X_2 + 0.20X_1 * X_3 - 0.029X_2 * X_3$; where 0.81, 11.53 and 4.05 are the absolute values of the first term of the equation, respectively. The results showed that the yield of *L. japonica* was affected by the single factor of N, P, and K in the order of P>K>N. Indicating that it can be seen that P fertilizer had the greatest impact on the yield of honeysuckle, while N fertilizer had the smallest effect. This result was consistent with results in Table 2.

The correlation coefficient $R^2$ was 0.9938 ($P=0.0018<0.05$) obtained by using DPS software for multiple regression analysis and test. It passed F-test and reached a significant level. The results indicated a high correlation between the independent variables, x and y (yield). Furthermore, a significant regression relationship was observed for the yield of *L. japonica* in response to the application of N, P, and K fertilizers. However, the coefficient of the first term in the equation exhibited both positive and negative values, which deviated from the conventional ternary quadratic fertilizer effect function.The calculation of recommended fertilization amount by using the marginal derivative method was not consistent with actual production. Therefore, frequency analysis method is used to calculate the recommended fertilization amount.

Analysis of single factor fertilizer effect function Single factor refers to fixing the fertilizer level of two factors and the relationship equation between fertilizer change and yield change was fitted. From Table 2, it can be seen that the fitting of the single factor fertilizer effect function was the same as that of the 3 factors, and the fertilizer usage and yield of N, P, and K fertilizers were applied for fitting. Regression analysis was conducted on N fertilizer effect function and correlation coefficients with F and P values using treatments N numbered 2, 3, 6, and 11 in Table 1. Similarly, the effect equations of P fertilizer were fitted using treatments 4, 5, 6, and 7, while the effect equations of K fertilizer were fitted using treatments 6, 8, 9, and 10. The effect equations of each fertilizer are detailed in Table 2. The results showed that the correlation coefficient R of the fertilizer effect function of N fertilizer was 0.9868 ($P=0.1621>0.05$), and the regression effect of the equation was not significant. The correlation coefficient R of the fertilizer effect function of P fertilizer was 0.9955, ($P=0.0944>0.05$), and the regression effect of the equation is not significant. The correlation coefficient R of the fertilizer effect function of K fertilizer is 0.9990 ($P=0.0440<0.05$), and the regression effect of the equation is significant.

From Table 4 and Table 5, it can be seen that the maximum fertilization amounts of N, $P_2O_5$, and $K_2O$ for each factor in the single factor fertilizer effect function are 7.5, 13, and 19.29, respectively. The corresponding maximum fertilization amounts are 408.45 g/plant, 409.03 g/plant, and 445.99 g/plant, respectively.

**Analysis of two-factor fertilizer effect function.** The two-factor fertilizer equation was an equation that fixes level of one fertilizer unchanged and fits the changes in the levels of two fertilizers with the changes in yield. The fitting of the

**Table 4. Analysis on single factor fertilizer effect function.**

| Factor | Fertilizer effect function | $R^2$ | P |
|---|---|---|---|
| N fertilizer | $Y = 340.95 + 3.60X_1 - 0.048X_1^2$ | 0.9868 | 0.1621 |
| P fertilizer | $Y = 316.55 + 14.31X_2 - 0.55X_2^2$ | 0.9955 | 0.0944 |
| K fertilizer | $Y = 322.40 + 8.11X_3 - 0.21X_3^2$ | 0.9990 | 0.0440 |

https://doi.org/10.6084/m9.figshare.28826912.

interaction effect function between the two factors was the same as that of the three factors. From Table 6, it can be seen that the regression analysis was performed on the fertilizer dosage and yield through NP, NK, and PK fertilizer related treatments. The interaction effect equation of NP fertilizer was obtained through regression analysis between the fertilizer dosage and yield of treatments numbered 2, 3, 4, 5, 6, 7, 11, and 12 in Table 1. Similarly, the interaction effect equation of NK fertilizer was obtained through regression analysis of fertilizer dosage 2 and yield under treatments of 2, 3, 6, 8, 9, 10, 11, and 13. The interaction effect equation of PK fertilizer was obtained through regression analysis of fertilizer dosage 2 and yield under treatments of 4, 5, 6, 7, 8, 9, 10, and 14. The results showed that the correlation coefficient $R^2$ of the fertilizer effect function of NP fertilizer was 0.9946, $P=0.0269<0.05$, indicating a significant regression effect of the equation; The correlation coefficient $R^2$ of the fertilizer effect function of NK fertilizer was 0.9898, $P=0.0498<0.05$, and the regression effect of the equation was significant; The correlation coefficient $R^2$ of the fertilizer effect function of PK fertilizer was 0.9948, $P=0.0259<0.05$, and the equation regression effect was significant.

**Production frequency analysis.** According to the above analysis, the regression effects of N and P fertilizers were not significant in the single factor fertilizer effect function, while the regression effects of K fertilizers were significant. The regression effects of NP, NK, and PK fertilizers were significant in the fertilizer effect functions of the interaction of two factors. The ternary quadratic fertilizer effect function was not a typical ternary quadratic fertilizer effect function. Therefore, this study was not suitable for using the marginal derivative method to calculate the maximum yield fertilization amount and the optimal fertilization amount. For results that did not fit the typical ternary quadratic fertilizer effect function, the yield frequency analysis method could be used to determine the appropriate recommended fertilization amount.

**Setting statistical intervals (Yield Range).** According to the results of the field experiment (Table 7), the yield range was set. The maximum actual average yield of this experimental community was 409 g/plant, and the minimum was 296 g/plant. The yield range (statistical interval) was set to $296<y<409$. Frequency analysis was calculated based on the field yield statistical table of *L. japonica*.

**Yield frequency analysis.** The frequency of each factor level of nitrogen, phosphorus and potassium falling into the set yield range was counted respectively, and the average, standard deviation, and 95% confidence interval recommended fertilization amount of nitrogen, phosphorus, and potassium according to formulas (1–4) were calculated to obtain the yield frequency analysis of honeysuckle, as shown in Table 8. The frequency of nitrogen fertilizer yields

Table 5. Single factor fertilizer effect function each factor fertilizer application and maximum yield analysis.

| Factor functions | Maximum fertilization amount of each factor (g/plant) | Maximum output (g/plant) |
|---|---|---|
| Fertilizer effect function of N | 7.5 | 408.45 |
| Fertilizer effect function of P | 13 | 409.03 |
| Fertilizer effect function of K | 19.29 | 445.99 |

https://doi.org/10.6084/m9.figshare.28826912.

Table 6. Analysis on fertilizer effect function of two factors.

| Factor | Fertilizer effect function | $R^2$ | P |
|---|---|---|---|
| NP fertilizer | $Y=303.10+1.94X_1+9.72X_2-0.051X_1^2-0.60X_2^2+0.18X_1*X_2$ | 0.9946 | 0.0269 |
| NK fertilizer | $Y=382.48-0.80X_1+1.35X_3-0.042X_1^2-0.24X_3^2+0.25X_1*X_3$ | 0.9898 | 0.0498 |
| PK fertilizer | $Y=260.41+11.11X_2+7.36X_3-0.50X_2^2-0.24X_3^2+0.15X_2*X_3$ | 0.9948 | 0.0259 |

https://doi.org/10.6084/m9.figshare.28826912.

**Table 7. Field trial yield of *L. japonica*.**

| Number | Treatment | Nutrient consumption (g/plant) | | | Yield per plant (g/plant) | | | |
|---|---|---|---|---|---|---|---|---|
| | | N | $P_2O_5$ | $K_2O$ | I | II | III | Average value |
| 1 | $N_0P_0K_0$ | 0 | 0 | 0 | 297 | 296 | 295 | 296 |
| 2 | $N_0P_2K_2$ | 0 | 10 | 16 | 337 | 338 | 340 | 339 |
| 3 | $N_1P_2K_2$ | 15 | 10 | 16 | 386 | 396 | 387 | 390 |
| 4 | $N_2P_0K_2$ | 30 | 0 | 16 | 313 | 307 | 325 | 315 |
| 5 | $N_2P_1K_2$ | 30 | 5 | 16 | 376 | 370 | 390 | 379 |
| 6 | $N_2P_2K_2$ | 30 | 10 | 16 | 391 | 402 | 407 | 400 |
| 7 | $N_2P_3K_2$ | 30 | 15 | 16 | 406 | 412 | 408 | 409 |
| 8 | $N_2P_2K_0$ | 30 | 10 | 0 | 324 | 323 | 322 | 323 |
| 9 | $N_2P_2K_1$ | 30 | 10 | 8 | 370 | 369 | 378 | 372 |
| 10 | $N_2P_2K_3$ | 30 | 10 | 24 | 395 | 397 | 394 | 395 |
| 11 | $N_3P_2K_2$ | 45 | 10 | 16 | 407 | 413 | 405 | 408 |
| 12 | $N_1P_1K_2$ | 15 | 5 | 16 | 364 | 373 | 366 | 368 |
| 13 | $N_1P_2K_1$ | 15 | 10 | 8 | 390 | 381 | 390 | 387 |
| 14 | $N_2P_1K_1$ | 30 | 5 | 8 | 359 | 341 | 360 | 353 |

https://doi.org/10.6084/m9.figshare.28826912.

**Table 8. Frequency analysis of *L. japonica* output.**

| Fertilization level | N fertilization amount | | | P fertilization amount | | | K fertilization amount | | |
|---|---|---|---|---|---|---|---|---|---|
| | Fertilization amount (g/plant) | Frequency | Rate(%) | Fertilization amount (g/plant) | Frequency | Rate (%) | Fertilization amount (g/plant) | Frequency | Rate (%) |
| 0 | 0 | 5 | 13.51 | 0 | 5 | 12.20 | 0 | 5 | 12.20 |
| 1 | 15 | 9 | 24.32 | 5 | 9 | 21.95 | 8 | 9 | 21.95 |
| 2 | 30 | 21 | 56.76 | 10 | 24 | 58.54 | 16 | 24 | 58.54 |
| 3 | 45 | 2 | 5.41 | 15 | 3 | 7.32 | 24 | 3 | 7.32 |
| Sum of times | | 37 | | | 41 | | | 41 | |
| $x_i$ average value | 22.5 | | | 7.5 | | | 12 | | |
| Standard deviation (s) | 13.17 | | | 4.05 | | | 6.40 | | |
| Standard deviation | 2.17 | | | 0.63 | | | 1.00 | | |
| 95%confidence interval | 18.25-26.75 | | | 6.27-8.73 | | | 10.04-13.96 | | |

https://doi.org/10.6084/m9.figshare.28826912.

falling within the designated interval at the 0–3 level was recorded as 5, 9, 21, and 2 times. Similarly, the occurrences of phosphorus fertilizer yields within the same interval were 5, 9, 24, and 3 times. For potassium fertilizer, the yields falling into the specified interval were also 5, 9, 24, and 3 times. The average application rates for nitrogen (N), phosphorus (P), and potassium (K) fertilizers were 23.57 g/plant, 7.86 g/plant, and 12.57 g/plant, respectively. The recommended application amounts for nitrogen fertilizer range from 18.25 to 26.75 g/plant, for phosphorus fertilizer ($P_2O_5$) from 6.27 to 8.73 g/plant, and for potassium fertilizer ($K_2O$) from 10.04 to 13.96 g/plant, which correspond to a yield range of 376.45 to 397.94 g/plant.

**Calculation and optimization of fertilization rate and yield.** The average application rates of nitrogen, phosphorus, and potassium fertilizers obtained by frequency analysis was 22.5 g/plant, 7.5 g/plant, and 12 g/plant, which were brought into the ternary quadratic fertilizer effect function, and the optimized fertilization yield was calculated to be 386.95 g/plant.

## Discussion

Nitrogen, phosphorus, and potassium are the three primary nutrient elements essential for plant growth and development, exhibiting interactive effects within the plant system. Nitrogen is crucial for the synthesis of proteins, nucleic acids, and chlorophyll, which promotes photosynthesis and enhances both the yield and quality of crops. Phosphorus, an essential component of nucleic acids and nuclear proteins, enhances the number of buds and the thousand-grain weight, ultimately contributing to improved yields in honeysuckle. Potassium absorbed by plant roots in ionic form, plays a crucial role in fundamental metabolic processes, thereby influencing plant growth and development while enhancing crop stress resistance. The application of nitrogen, phosphorus, and potassium fertilizers to the soil facilitates interactions among various nutrient elements and soil microorganisms, thereby promoting a synergistic enhancement of their effects. This interaction maximizes the efficacy of these fertilizers, ultimately improving their utilization rates and enhancing both crop yield and quality [28,29]. In this study, the yield of *L. japonica* was observed to increase with the rising levels of nitrogen and phosphorus content. Inorganic elements, specifically nitrogen (N), phosphorus (P), and potassium (K), represent the mineral nutrients that are most essential for plant metabolism, the formation of tissues and organs, and overall growth and development. Fertilization has been utilized in agricultural production to meet the essential nutrient requirements of plants, specifically nitrogen (N), phosphorus (P), and potassium (K) [30]. Cao et al. [31] demonstrated that balanced fertilization with N, P, and K significantly enhances plant growth and development, as evidenced by increases in the number of flower nodes, fresh weight of flowers, and the number of flower buds, ultimately leading to flower bud yield. Research has demonstrated that fertilization can enhance the yield of *L. japonica*. As fertilization levels increase, the yield per plant also rises. However, when the amount of fertilization exceeds a certain threshold, there is a significant decline in yield [32,33]. A single experiment conducted on nitrogen (N), phosphorus (P), and potassium (K) demonstrated that their effectiveness in enhancing the yield of L. japonica follows the order: $N > P > K$.

The optimal application rates for these nutrients were determined to be 0.1 kg/plant for N, 0.2 kg/plant for P, and 0.1 kg/plant for K [34]. The yield study results indicated that N, P, K, and compound fertilizers significantly enhance the differentiation of flower buds in honeysuckle plants, which is positively correlated with the length of the flower buds. Among these elements, N has the potential to enhance the yield of both leaves and flowers. In contrast, P serves as the primary fertility factor influencing flower bud weight and yield, playing a crucial role in the growth and development of flower buds [35]. In this study, varying fertilizer ratios significantly influenced the yield of *L. japonica* per plant. Specifically, the yield for the treatment $N_2P_3K_2$ increased by 38.18% compared to the control group. Additionally, the contribution rate of basic soil fertility at the field trial site was determined to be 73.97%. Within the recommended range for fertilization range, the application of 1 g of nitrogen, phosphorus, and potassium fertilizer has the potential to increase the yield of *L. japonica* buds by 2.04 g, 8.52 g, and 4.82 g, respectively.

The results indicated that P fertilizer was the primary limiting factor in enhancing the yield of *L. japonica* buds. Therefore, it is essential to consider the quantity of P fertilizer applied to achieve high yields, a finding that aligns with the results reported by Zeng Hui jie et al [30]. Tongwei County is situated in the hilly and gully region of the Loess Plateau in central Gansu. The correlation coefficient ($R^2$) for the equation representing the fertilizer effect of *L. japonica* was determined through regression analysis. This analysis was based on the yield of *L. japonica* obtained from experiments, as well as the quantities of N, P, and K. The resulting $R^2$ value was 0.9938, with a p-value of 0.0018, which is less than 0.05, indicating a statistically significant regression effect. However, the coefficients of the first terms were -0.81, 11.53, and 4.05, respectively. These values represent typical fertilizer effect functions that do not adhere conform to the law of diminishing returns. The F-test produced a p-value of 0.0018, which is below the significance threshold of 0.05, thereby indicating

a statistically significant result. Additionally, the marginal analysis method was employed to determine both the optimal fertilization amount and the maximum yield fertilization amount. However, these findings derived from this analysis were inconsistent with actual production levels. The recommended fertilization amount for 4-year-old *L. japonica*, as determined through frequency analysis, is as follows: N fertilizer should be applied at a rate of 18.25 to 26.75 g/plant. The recommended application rate for P fertilizer ($P_2O_5$) is 6.27 to 8.73 g/plant, while the suggested range for K fertilizer ($K_2O$) is from 10.04 to 13.96 g/plant. The anticipated yield per plant is projected to be between 376.45 grams and 397.94 grams. The conventional fertilization amounts, based on local yield levels and data from Tongwei County spanning 2016–2021, were established at 30 g of pure nitrogen per plant, 10 g of $P_2O_5$ per plant, and 16 g of $K_2O$ per plant. The observed amounts exceeded the recommended fertilization levels established through frequency analysis. Therefore, it is advisable to adjust the recommended fertilization amount for the upcoming year's tests to a slightly lower level. This method has been applied to *Scutellaria Radix, Paeoniae Radix Alba,* and *Uncaria Rhynchophylla*. These findings can also be extended to certain varieties, such as Four-season *Honeysuckle* from Shandong Province, Juhua No. 1 from Hebei Province, and Jiufeng No. 1 from Hubei Province [36–39].

In summary, due to variations in soil texture, and fertility, fertilization strategies should be tailored to the specific conditions of each planting region. Furthermore, it is essential to ensure a balanced application of inorganic fertilizers to prevent soil compaction and imbalances in microbial flora, thereby promoting the development of the *L. japonica* industry.

## Conclusion

This study demonstrates that the application of formula fertilization with varying ratios of nitrogen (N), phosphorus (P), and potassium (K) significantly enhances the yield of fresh buds of L. japonica in regions of China characterized by poor soil nutrient availability. The yield results fit a concise quadratic equation: $Y=296.66-0.81X_1+11.53X_2+4.05X_3-0.051X_1^2-0.60X_2^2-0.28X_3^2+0.13X_1 \times X_2+0.20X_1 \times X_3-0.029X_2 \times X_3$. Frequency analysis indicates that the recommended fertilizer dosages are 18.25 to 26.75 g/plant for N, 6.27 to 8.73 g/plant for $P_2O_5$, and 10.04 to 13.96 g/plant for $K_2O$.

## Author contributions

**Conceptualization:** Xiaofeng Jiang, Yusheng Zhai.

**Data curation:** Xiaofeng Jiang, Jianguo Guo, Shangzhong Li, Yusheng Zhai.

**Formal analysis:** Yuan Chen, Yusheng Zhai.

**Funding acquisition:** Xiaofeng Jiang, Yusheng Zhai.

**Investigation:** Xiaofeng Jiang, Ping Xu, Bo Dong.

**Methodology:** Yuan Chen, Shangzhong Li.

**Resources:** Jianguo Guo, Yuan Chen, Zhaohui Shi.

**Supervision:** Xiaofeng Jiang.

**Validation:** Shangzhong Li.

**Visualization:** Zhaohui Shi, Ping Xu.

**Writing – original draft:** Xiaofeng Jiang, Jiali Chen.

**Writing – review & editing:** Xiaofeng Jiang, Jiali Chen, Qing Fang.

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
