## [Decision Letter · Decision Letter 0]

2 Apr 2024

PONE-D-24-06358Study on Nitrogen, Phosphorus and Potassium Formula Fertilization Model of Lonicera japonica Thunb in hilly and gully Loess Plateau of ChinaPLOS ONE

Dear Dr. Jiang,

Thank you for submitting your manuscript to PLOS ONE. After careful consideration, we feel that it has merit but does not fully meet PLOS ONE’s publication criteria as it currently stands. Therefore, we invite you to submit a revised version of the manuscript that addresses the points raised during the review process.

We look forward to receiving your revised manuscript.

Kind regards,

Susmita Lahiri (Ganguly)

Academic Editor

PLOS ONE

Journal Requirements:

"National key research and development project and sub-project (2021YFD1100504�2021YFD1100504-1�2021YFD1100504-2). Agricultural Science and technology independent innovation special key research and development program of Gansu Academy of Agricultural Sciences(2021GARS11); Gansu Province Science and Technology Plan project: Livelihood Science and technology special - Topics on rural revitalization(22CX2NA007); National Natural Science Foundation of China (NSFC)(31460547)"

"no"

Reviewers' comments:

Reviewer's Responses to Questions

**Comments to the Author**

1. Is the manuscript technically sound, and do the data support the conclusions?

Reviewer #1: Yes

Reviewer #2: Yes

Reviewer #3: Yes

2. Has the statistical analysis been performed appropriately and rigorously? 

Reviewer #1: Yes

Reviewer #2: Yes

Reviewer #3: Yes

3. Have the authors made all data underlying the findings in their manuscript fully available?

Reviewer #1: Yes

Reviewer #2: Yes

Reviewer #3: Yes

4. Is the manuscript presented in an intelligible fashion and written in standard English?

Reviewer #1: Yes

Reviewer #2: Yes

Reviewer #3: Yes

5. Review Comments to the Author

Reviewer #1: The abstract section does not provide an overview of the research methods

The methods section does not include how to administer NPK, watering techniques and pest control.

Watering techniques or providing air to plants are still ignored in the discussion.

The conclusion needs to be adjusted to the problem formulation

Reviewer #2: The study presented in this paper investigates the effects of various fertilization treatments on the yield and economic benefits of Lonicera japonica (L. japonica). The authors conducted a comprehensive field trial, analyzed the relative yield and increasing rate of fertilizer deficiency treatments, performed regression analysis for fertilizer effect functions, and conducted yield frequency analysis to determine the optimal fertilization rates for maximizing yield.

The paper begins by introducing the concept of soil nutrient abundance and deficiency index, which is crucial for understanding the impact of different fertilization treatments on crop yield. The authors categorize soil nutrients based on their relative yield percentages and establish a correlation between soil fertility and the yield of L. japonica. Notably, the results highlight phosphorus (P) as the primary fertility factor limiting the yield of L. japonica, indicating the importance of balanced fertilization strategies. The paper also delves into the analysis of fertilizer effect functions, including ternary quadratic and single-factor functions. Through regression analysis and correlation coefficients, the authors identify the relative impact of N, P, and K fertilizers on L. japonica yield. The study concludes with yield frequency analysis, which calculates recommended fertilization amounts based on statistical intervals, optimizing fertilization rates for maximizing yield.

Overall, the paper presents a well-structured and scientifically rigorous analysis of fertilization treatments' effects on L. japonica yield and economic benefits. The findings contribute significantly to agricultural practices, particularly in optimizing fertilization strategies for enhancing crop productivity and profitability.

Reviewer #3: Remarks to the Author and editor

Manuscript Number PONE-D-24-06358

Title: Study on Nitrogen, Phosphorus and Potassium Formula Fertilization Model of Lonicera japonica Thunb in hilly and gully Loess Plateau of China

General comments

The present study was conducted to the aim of this study is to establish models for higher yield of L. japonica by formula fertilization with different ratios of nitrogen (N), phosphorus (P) and potassium (K) in main producing areas in hilly and gully Loess Plateau of China. A ‘3414’ formula fertilization trial with different dose and ratios of N, P, and K used for L. japonica was investigated. The results showed the application of N, P and K content in soil improved soil nutrient effectively. And P was identified as the primary limiting factor for yield. The ratio of output to investment of N2P2K3 was lowered significantly. The fertilizer effect function was: Y = 296.66 - 0.81X1 + 11.53X2 + 4.05X3 - 0.051X12 - 0.60X22 - 0.28X32 + 0.13X1×X2 + 0.20X1×X3 - 0.029X2×X3. The recommended dose of N, P, and K fertilizer was 18.25-26.75 g, 6.27-8.73 g, and 10.04-13.96 g, respectively, per plant. By applying the fertilizer with formula from this research, the yield and economic benefit increased notably by 38.18% and 2.71 CNY.

The objectives of the study are clear, methodology is understandable, and discussion is well written, apart from that, there are a few suggestions for the improvement of this paper, which are as followed

1. In abstracts some figurative result outcomes should be further elaborated.

2. The number of keywords should be at least five.

3. Gansu Province is a main producing area in China, of what? please clarify the statement.

4. At Line number 70, s should be small in Sunshine.

5. Use proper language, at line number 86 replace the word “test site” with experimental site and at 87 test began, as experiment began.

6. Avoid unnecessary capitalization of words for example at line number 89 “Continental” The Times at line number 281 and 283.

7. In material and method section, please describe the soil preparation for cultivation of L. japonica species, irrigation and agronomic practices adopted for this experiment.

8. Line number 126 Yield measurement, which yield parameters were recorded, please elaborate.

9. Line number 153, Do not use bracket in the beginning of the sentence as you did (It is high to determine the soil nutrient abundance and deficiency index of a certain crop in a certain area and the corresponding fertilization amount.).

10. Line number 163, potassium is on an average degrade? Please check this sentence.

11. How economic benefit was calculated please describe a formula or method in material & method section.

12. Increase number of references, add them at appropriate places.

6. PLOS authors have the option to publish the peer review history of their article (what does this mean? ). If published, this will include your full peer review and any attached files.

**Do you want your identity to be public for this peer review?** For information about this choice, including consent withdrawal, please see our Privacy Policy .

Reviewer #1: **Yes: ** I Nengah Muliarta

Reviewer #2: No

Reviewer #3: No

---

## [Author Response · Author response to Decision Letter 1]

23 May 2024

Response to Reviewer Comments

Dear Editors and Reviewers:

We are very grateful for the further comments concerning our manuscript entitled “Study on Nitrogen, Phosphorus and Potassium Formula Fertilization Model of　Lonicera japonica Thunb in hilly and gully Loess Plateau of China (PONE-D-24-06358)”. These comments are all valuable and very helpful for the further revision and improvement of our article, which also provides an important guideline for our researches and scientific writing. We have carefully studied the comments and made corrections that we hope meet the standards for publication. If you have any questions regarding this manuscript, please do not hesitate to contact me. The detailed corrections in the article and the responses to the editors' and reviewers' comments are as follows:

Response to Editor Requirements:

“When submitting your revision, we need you to address these additional requirements.”

Answer: The paper has been modified according to The PLOS ONE style templates.

Answer: The experimental site for this experiment was an authorized experimental base of Gansu Agricultural Academy, which has been approved and authorized by the provincial government to serve as an experimental field site. And we have corrected as you required. (Line 100-102)

3. Thank you for stating the following financial disclosure: "National key research and development project and sub-project (2021YFD1100504�2021YFD1100504-1�2021YFD1100504-2). Agricultural Science and technology independent innovation special key research and development program of Gansu Academy of Agricultural Sciences(2021GARS11); Gansu Province Science and Technology Plan project: Livelihood Science and technology special - Topics on rural revitalization(22CX2NA007); National Natural Science Foundation of China (NSFC)(31460547)" Please state what role the funders took in the study. If the funders had no role, please state: "The funders had no role in study design, data collection and analysis, decision to publish, or preparation of the manuscript." If this statement is not correct you must amend it as needed. Please include this amended Role of Funder statement in your cover letter; we will change the online submission form on your behalf.

Answer: The National Key Research and Development Project and its sub-projects (2021YFD1100504, 2021YFD1100504-1, 2021YFD1100504-2) are national-level scientific research projects on honeysuckle currently being carried out by the project team. These three projects have made significant contributions in terms of personnel allocation, technical support, and financial support.The Agricultural Science and technology independent innovation special key research and development program of Gansu Academy of Agricultural Sciences (2021GARS11) provided Sharing of technical exchanges and experimental bases. (Line 409-416)

"no" Please complete your Competing Interests on the online submission form to state any Competing Interests. If you have no competing interests, please state "The authors have declared that no competing interests exist.", as detailed online in our guide for authors at http://journals.plos.org/plosone/s/submit-now

Answer: The authors have declared that no competing interests exist, and have changed the online submission form as you required.

Answer: The authors have registered an account as you required.

Response to the Reviewer#

1. The abstract section does not provide an overview of the research methods

Answer: we have modified as required. (Line 29-30)

2. The number of keywords should be at least five.

Answer: we have modified as required.

3. Gansu Province is a main producing area in China, of what? please clarify the statement.

Answer: Gansu Province is the main producing area of most authentic Chinese medicinal materials in China. (Line 46-47)

4. At Line number 70, s should be small in Sunshine.

Answer: we have modified as required. (Line 72)

5. Use proper language, at line number 86 replace the word “test site” with experimental site and at 87 test began, as experiment began.

Answer: we have modified as required. (Line89)

6. Avoid unnecessary capitalization of words for example at line number 89 “Continental” The Times at line number 281 and 283.

Answer: we have modified as required. (Line 306-308)

7. In material and method section, please describe the soil preparation for cultivation of L. japonica species, irrigation and agronomic practices adopted for this experiment.

Answer: we have modified as required. The experiment located in Tongwei, Gansu Province, the temperate belongs to semi-humid and semi-arid monsoon climate type. Therefore, the irrigation only depended on rainfall, the water, light and other natural factors are consistent. Artificial management of weeding and uniform spraying of pesticides were used during growth period. 24g of 25% imidacloprid wettable powder with 1000 fold solution were spray per hectare for the plant. (Line 96-99).

The formulized fertilizer was mixed evenly, and the fertilizer were poured into a small pit of 30 cm deep around the root of each tree. (Line 120-121).

8. Line number 126 Yield measurement, which yield parameters were recorded, please elaborate.

Answer: we have modified as required. The yield measurement was through the manual picking of workers, each person picked six trees, the flower weight of each tree was weighed with a high-precision electronic balance (model AE001, Jiangsu Hongli weighing Equipment Co., LTD.), and the yield of each tree was recorded. The average value of the flower yield of six trees was the yield of a single tree. (Line 137-141)

9. Line number 153, Do not use bracket in the beginning of the sentence as you did (It is high to determine the soil nutrient abundance and deficiency index of a certain crop in a certain area and the corresponding fertilization amount.)

Answer: we have modified as required. (Line 174-176)

10. Line number 163, potassium is on an average degrade? Please check this sentence.

Answer: we have modified as required. (Line182-184)

11. How economic benefit was calculated please describe a formula or method in material & method section.

Answer: we have described the information as you required. (Line159-164)

The economic benefits were calculated as follows:

Output value (CNY/plant) = Yield of per plant*24/1000 (5)

Fertilizer input (CNY/plant) = (nitrogen fertilizer amount of per plant *4.8/1000) + (phosphorus fertilizer amount of per plant *6/1000) + (potassium fertilizer amount of per plant *8/1000)

Yield to investment ratio (%) = output value /fertilizer input (7)

12. Increase number of references, add them at appropriate places.

Answer: we have corrected as you required.

Response to the Reviewer#

1. The abstract provides a clear overview of the study's objectives and methods. However, there are a few grammatical issues that need to be addressed, such as:

"Formula fertilization is effective in increasing yield and economic profit in medicinal plants, but no study on performance on Lonicera japonica Thunb ( L. japonica) has been conducted, which is an important medicinal plant." - This sentence is a bit convoluted. It could be rephrased for better clarity.

"The results showed the application of N, P and K content in soil improved soil nutrient effectively." - It would be helpful to specify how the soil nutrient content was improved by the application of N, P, and K fertilizers.

Answer: we have corrected as you required. (Line 23-24, 32)

2. The introduction provides a good background on Lonicera japonica and its medicinal properties. However, some areas require further explanation or elaboration. The statement "Formula fertilization is an effective way to improve yield and quality of L. japonica" could be supported with references or previous studies demonstrating this effectiveness. It would be beneficial to include more details about the formula fertilization trial mentioned in the introduction, such as the specific dosages and ratios of N, P, and K used. The paragraph discussing the geographical conditions and cultivation practices in Tongwei County could be expanded to include information on climate, soil types, and any unique challenges faced in cultivating L. japonica. Overall, the abstract and introduction provide a solid foundation for the study, but addressing these points would enhance the clarity and comprehensiveness of the paper.

Answer: we have corrected as you required. (Line 53,58,96-99)

3. Table 2 provides a detailed overview of the relative yield and increasing rate of fertilizer deficiency treatments, demonstrating the incremental yield associated with nitrogen (N), phosphorus (P2O5), and potassium (K2O) fertilizers. The data suggest that increasing P fertilizer has a significant positive effect on improving the yield of L. japonica buds, aligning with the earlier observation regarding the importance of P in soil fertility. Can you provide more details about the methodology used to determine the relative yield and increasing rate of fertilizer deficiency treatments presented in Table 2? How were the yield values in Table 2 calculated, particularly for the different fertilization treatments (N0P0K0, N0P2K2, N2P0K2, etc.)?

Answer: The yield measurement was through the manual picking of workers, each person picked six trees, the flower weight of each tree was weighed with a high-precision electronic balance. The yield of each tree was recorded. The average value of the flower yield of six trees was the yield of a single tree. (Line 136-141).

4. Table 3 showcases the effects of different treatments on L. japonica yield and economic benefits. The authors discuss the yield-per-plant variations across various fertilization treatments, emphasizing the substantial increase in yield and economic profit with optimized fertilization strategies. The analysis of output-to-investment ratios provides valuable insights into the cost-effectiveness of different fertilization regimes, with treatments like N1P2K1 exhibiting a notably high output ratio. What specific factors were considered when analyzing the economic benefits in Table 3? Did the analysis include factors such as market prices, production costs, or other economic variables? Can you elaborate on the criteria used to determine the highest output-to-investment ratio in Table 3? How does this ratio contribute to the economic assessment of different fertilization treatments?

Answer:

The frequency analysis calculation formula was:

(1)

In the formula, nij represents the number of occurrences of the variable value Xij, where (uj), and nj represents the total number of occurrences of the variable Xi.

Standard deviation:

(2)

Mean standard deviation:

(3)

95% confidence interval

(4)

The economic benefits were calculated as follows:

Output value (CNY/plant) = Yield of per plant*24/1000 (5)

Fertilizer input (CNY/plant) = (nitrogen fertilizer amount of per plant *4.8/1000) + (phosphorus fertilizer amount of per plant *6/1000) + (potassium fertilizer amount of per plant *8/1000) (6)

Yield to investment ratio (%)= output value /fertilizer input (7)

5. The discussion effectively highlights the importance of N, P, and K fertilization in promoting plant growth and development. However, it would be beneficial to provide more specific details on how these elements interact and contribute to yield improvement in Lonicera japonica. For example, what mechanisms are involved in promoting flower bud differentiation and yield increase with N, P, and K fertilizers? � The discussion mentions the impact of different fertilizer ratios on the yield of L. japonica and presents recommended fertilization amounts. It's important to elaborate on how these ratios and amounts were determined. Was there a specific methodology or experimental design used to establish these recommendations? Additionally, it would be helpful to discuss any potential limitations or challenges in implementing these recommended fertilization amounts, especially considering variations in soil conditions and environmental factors across different planting regions. The discussion briefly compares the findings of this study with previous research, which is good for contextualizing the results. However, it could be expanded by discussing any discrepancies or similarities in results between this study and others in the field.

Answer: Thank you for your valuable suggestions, we have corrected as you required. (Line 322-334)

---

## [Decision Letter · Decision Letter 1]

4 Jun 2024

PONE-D-24-06358R1Study on Nitrogen, Phosphorus and Potassium Formula Fertilization Model of Lonicera japonicaThunb in hilly and gully Loess Plateau of ChinaPLOS ONE

Dear Dr. Jiang,

Thank you for submitting your manuscript to PLOS ONE. After careful consideration, we feel that it has merit but does not fully meet PLOS ONE’s publication criteria as it currently stands. Therefore, we invite you to submit a revised version of the manuscript that addresses the points raised during the review process.

We look forward to receiving your revised manuscript.

Kind regards,

Susmita Lahiri (Ganguly)

Academic Editor

PLOS ONE

Journal Requirements:

Reviewers' comments:

Reviewer's Responses to Questions

**Comments to the Author**

1. If the authors have adequately addressed your comments raised in a previous round of review and you feel that this manuscript is now acceptable for publication, you may indicate that here to bypass the “Comments to the Author” section, enter your conflict of interest statement in the “Confidential to Editor” section, and submit your "Accept" recommendation.

Reviewer #1: All comments have been addressed

Reviewer #3: All comments have been addressed

2. Is the manuscript technically sound, and do the data support the conclusions?

Reviewer #1: Yes

Reviewer #3: Yes

3. Has the statistical analysis been performed appropriately and rigorously? 

Reviewer #1: Yes

Reviewer #3: Yes

4. Have the authors made all data underlying the findings in their manuscript fully available?

Reviewer #1: Yes

Reviewer #3: Yes

5. Is the manuscript presented in an intelligible fashion and written in standard English?

Reviewer #1: Yes

Reviewer #3: Yes

6. Review Comments to the Author

Reviewer #1: 1. pay attention to writing Latin names, such as the family name in line 46 (Caprifoliaceae)

2. Lines 72-73 where is the data source? please include it

3. pay attention to the writing on line 129

4. provide an analysis of the statements in lines 168-172. What caused it?

5. reference number 12 line 456 is quite old, can you find a newer reference?

6. What year is reference number 20 line 486?

Reviewer #3: The authors has made significant improvement on revision, hence I have no hesitation to accept this manuscript for publication.

7. PLOS authors have the option to publish the peer review history of their article (what does this mean? ). If published, this will include your full peer review and any attached files.

**Do you want your identity to be public for this peer review?** For information about this choice, including consent withdrawal, please see our Privacy Policy .

Reviewer #1: No

Reviewer #3: No

---

## [Author Response · Author response to Decision Letter 2]

4 Jul 2024

Response to Reviewer Comments

Dear Editors and Reviewers:

We are very grateful for the further comments concerning our manuscript entitled “Study on Nitrogen, Phosphorus and Potassium Formula Fertilization Model of　Lonicera japonica Thunb in hilly and gully Loess Plateau of China (PONE-D-24-06358)”. These comments are all valuable and very helpful for the further revision and improvement of our article, which also provides an important guideline for our researches and scientific writing. We have carefully studied the comments and made corrections that we hope meet the standards for publication. If you have any questions regarding this manuscript, please do not hesitate to contact me. The detailed corrections in the article and the responses to the editors' and reviewers' comments are as follows:

Response to Editor Requirements:

1. pay attention to writing Latin names, such as the family name in line 46 (Caprifoliaceae)

Answer: Thank you for your constructive suggestions! We have revised the manuscript as you required. (Line 46)

2.Lines 72-73 where is the data source? please include it

Answer: Thank you for your constructive suggestions! Tongwei is the place where the test is arranged, which is explained in the overview of the test site.

The data was from the reference of Niu Sh W,Wang Zh F, Li G Zh, Ma L B. Analyzing the Potentials and Structure of Rural Household Energy Use in Hilly Regions of the Loess Plateau�A Case Study in Qizuei Village of Tongwei County of Gansu Province. Resources Science. 2007,29(3):105-110. https://doi.org/1007-7588(2007)03-0105-06 (Line 73)

3.pay attention to the writing on line 129

Answer: Thank you for your constructive suggestions, and we feel terribly sorry for this mistake And we have revised as you required. (Line 129)

4. provide an analysis of the statements in lines 168-172. What caused it?

Answer: Thank you for your constructive suggestions! We have revised as required. (Line 173-185)

Answer:5. reference number 12 line 456 is quite old, can you find a newer reference?

Answer: Thank you for your constructive suggestions! We have revised as you required. Guo R,Liu R Y, Huang H Y, ZHai Y SH, CHen Y, Li W D. Effects of Different Fertilization Formulas on Yield and Quality of Lonicera japonica Flos and Soil Microecology in Loess Plateau.Journal of Nuclear Agricultural Sciences,2023,37(7):1452~1461. https://doi10.11869/j.issn.1000⁃8551.2023.07.1452.

6. What year is reference number 20 line 486?

Answer: Thank you for your constructive suggestions! We have revised as you required. (Line 494)

---

## [Decision Letter · Decision Letter 2]

3 Sep 2024

PONE-D-24-06358R2Study on Nitrogen, Phosphorus and Potassium Formula Fertilization Model of Lonicera japonicaThunb in hilly and gully Loess Plateau of ChinaPLOS ONE

Dear Dr. Jiang,

Thank you for submitting your manuscript to PLOS ONE. After careful consideration, we feel that it has merit but does not fully meet PLOS ONE’s publication criteria as it currently stands. Therefore, we invite you to submit a revised version of the manuscript that addresses the points raised during the review process.

We look forward to receiving your revised manuscript.

Kind regards,

Susmita Lahiri (Ganguly)

Academic Editor

PLOS ONE

**Journal Requirements:**

Reviewers' comments:

Reviewer's Responses to Questions

**Comments to the Author**

1. If the authors have adequately addressed your comments raised in a previous round of review and you feel that this manuscript is now acceptable for publication, you may indicate that here to bypass the “Comments to the Author” section, enter your conflict of interest statement in the “Confidential to Editor” section, and submit your "Accept" recommendation.

Reviewer #4: All comments have been addressed

Reviewer #5: (No Response)

2. Is the manuscript technically sound, and do the data support the conclusions?

Reviewer #4: Partly

Reviewer #5: Yes

3. Has the statistical analysis been performed appropriately and rigorously? 

Reviewer #4: Yes

Reviewer #5: Yes

4. Have the authors made all data underlying the findings in their manuscript fully available?

Reviewer #4: Yes

Reviewer #5: Yes

5. Is the manuscript presented in an intelligible fashion and written in standard English?

Reviewer #4: Yes

Reviewer #5: Yes

6. Review Comments to the Author

**Reviewer #4:**  The manuscript is well-prepared and I can see that the authors responded to some comments. as this is my first time to review this manuscript, I have noticed that the results section need to be revised. only present the results with no explanation or interpretation or comparison with previous work.

These can be done in the discussion section.

**Reviewer #5: ** The authors have well addressed the comments made previously. I recommend it accepting for publication after revising some minor corrections. In line (62-66) Could you please revise this long sentence into two clear sentences? This will make it easier to understand by clearly distinguishing between the different effects of nitrogen fertilizer.

Please follow the author guidelines for standard formatting of the manuscript.

7. PLOS authors have the option to publish the peer review history of their article (what does this mean? ). If published, this will include your full peer review and any attached files.

**Do you want your identity to be public for this peer review?** For information about this choice, including consent withdrawal, please see our Privacy Policy .

Reviewer #4: No

Reviewer #5: No

---

## [Author Response · Author response to Decision Letter 3]

27 Sep 2024

Response to Reviewer Comments

Dear Editors and Reviewers:

We are very grateful for the further comments concerning our manuscript entitled “Study on Nitrogen, Phosphorus and Potassium Formula Fertilization Model of　Lonicera japonica Thunb in hilly and gully Loess Plateau of China (PONE-D-24-06358)”. These comments are all valuable and very helpful for the further revision and improvement of our article, which also provides an important guideline for our researches and scientific writing. We have carefully studied the comments and made corrections that we hope meet the standards for publication. If you have any questions regarding this manuscript, please do not hesitate to contact me. The detailed corrections in the article and the responses to the editors' and reviewers' comments are as follows:

Response to Editor Requirements:

1.Journal Requirements:

Answer: I have carefully reviewed my paper references and they are complete and there are no references to papers that have been retracted.

2.Reviewer #4: The manuscript is well-prepared and I can see that the authors responded to some comments, as this is my first time to review this manuscript, I have noticed that the results section need to be revised. only present the results with no explanation or interpretation or comparison with previous work.

These can be done in the discussion section.

Answer: The discussion section has been modified in the original text according to the reviewer's suggestion.

3. Reviewer #5: The authors have well addressed the comments made previously. I recommend it accepting for publication after revising some minor corrections. In line (62-66) Could you please revise this long sentence into two clear sentences? This will make it easier to understand by clearly distinguishing between the different effects of nitrogen fertilizer.

Please follow the author guidelines for standard formatting of the manuscript.

Answer: Thank you for your valuable suggestions� We have corrected the manuscript as you required. “The application of nitrogen fertilizer in Japan enhanced the vegetative growth of L. japonica, increased the biological yield of vegetative organs, and elevated the crude protein content of the leaves. However, it resulted in a decrease in both the economic yield and the chlorogenic acid content of the buds of the reproductive organs. ”(Line62-67)

---

## [Decision Letter · Decision Letter 3]

13 Nov 2024

PONE-D-24-06358R3Study on Nitrogen, Phosphorus and Potassium Formula Fertilization Model of Lonicera japonicaThunb in hilly and gully Loess Plateau of ChinaPLOS ONE

Dear Dr. Jiang,

Thank you for submitting your manuscript to PLOS ONE. After careful consideration, we feel that it has merit but does not fully meet PLOS ONE’s publication criteria as it currently stands. Therefore, we invite you to submit a revised version of the manuscript that addresses the points raised during the review process.

We look forward to receiving your revised manuscript.

Kind regards,

Susmita Lahiri (Ganguly)

Academic Editor

PLOS ONE

Journal Requirements:

Reviewers' comments:

Reviewer's Responses to Questions

**Comments to the Author**

1. If the authors have adequately addressed your comments raised in a previous round of review and you feel that this manuscript is now acceptable for publication, you may indicate that here to bypass the “Comments to the Author” section, enter your conflict of interest statement in the “Confidential to Editor” section, and submit your "Accept" recommendation.

Reviewer #4: All comments have been addressed

Reviewer #6: All comments have been addressed

Reviewer #7: All comments have been addressed

2. Is the manuscript technically sound, and do the data support the conclusions?

Reviewer #4: Yes

Reviewer #6: Yes

Reviewer #7: Yes

3. Has the statistical analysis been performed appropriately and rigorously? 

Reviewer #4: Yes

Reviewer #6: Yes

Reviewer #7: Yes

4. Have the authors made all data underlying the findings in their manuscript fully available?

Reviewer #4: Yes

Reviewer #6: No

Reviewer #7: Yes

5. Is the manuscript presented in an intelligible fashion and written in standard English?

Reviewer #4: Yes

Reviewer #6: Yes

Reviewer #7: Yes

6. Review Comments to the Author

Reviewer #4: The author responded to the reviewers' comments. the manuscript cam be accepted if there is no negative comments from other reviwers.

Reviewer #6: Information is in the attachment file. I have put some corrections in the file named: RevisionPONE-D-24-06358R4

Reviewer #7: (No Response)

7. PLOS authors have the option to publish the peer review history of their article (what does this mean? ). If published, this will include your full peer review and any attached files.

**Do you want your identity to be public for this peer review?** For information about this choice, including consent withdrawal, please see our Privacy Policy .

Reviewer #4: No

Reviewer #6: No

Reviewer #7: No

---

## [Author Response · Author response to Decision Letter 4]

23 Dec 2024

Response to Reviewer Comments

Dear Editors and Reviewers:

We are very grateful for the further comments concerning our manuscript entitled “Study on Nitrogen, Phosphorus and Potassium Formula Fertilization Model of　Lonicera japonica Thunb in hilly and gully Loess Plateau of China (PONE-D-24-06358)”. These comments are all valuable and very helpful for the further revision and improvement of our article, which also provides an important guideline for our researches and scientific writing. We have carefully studied the comments and made corrections that we hope meet the standards for publication. If you have any questions regarding this manuscript, please do not hesitate to contact me. The detailed corrections in the article and the responses to the editors' and reviewers' comments are as follows:

Response to Editor Requirements:

1.Line 40. Please Change CNY by dollars or euros? Frequently, all international journal use USA dollars.

Answer: Based on Exchange rate between US Dollar and RMB on December 12, 2024:1USD≈7.2674CNY 1CNY≈ 0.1376USD

The yuan in the original text is completely revised to US dollars.(Line40)

2.Line 47. Remove Lonicera genus in the

Answer:“belongs to Lonicera genus in the Caprifoliaceae family” instead“belongs to the Caprifoliaceae family”(Line46-47)

3.Line 100. Please put soil texture. This information is essential to exchange cationic capacity.

Answer: The soil is loam。(Line101)

4.Line 103 is it grams of active ingredient? If yes. It should be g ai.Please consider this sentence: Imidacloprid wettable (25%), powder with 1000 fold solution, were spray at rate of 24 g per hectare for the plant.

Answer: yes. “Imidacloprid wettable (25%), powder with 1000 fold solution, were spray at rate of 24 g per hectare for the plant”instead“ Imidacloprid 25% wettable powder were sprayed at rate of 24 g a.i. per hectare for the plant”(line103-104)

5.108. Please put space in L. japonica

Line 113. Why do the authors not mention K in the title? And line 198 mention K

Answer: The original has been amended.(line108)

The author mentions potassium in the title.

6.122. I think 3 fertilizers and 4 levels involve 40 treatments. Why the research was done with only 14 treatments.

N0Q0P0 ; N0Q0P1; N0Q1P1 ; N0Q1P2 ; N0Q2P2 ; N0Q2P3 ; N0Q3P3 ; N0Q1P3; N0Q2P3; N0Q3P2 ; N1Q0P0 ; N1Q0P1; N1Q1P1 ; N1Q1P2 ; N1Q2P2 ; N1Q2P3 ; N1Q3P3 ; N1Q1P3; N1Q2P3; N1Q3P2 ; N2Q0P0 ; N2Q0P1; N2Q1P1 ; N2Q1P2 ; N2Q2P2 ; N2Q2P3 ; N2Q3P3 ; N2Q1P3; N2Q2P3; N2Q3P2 ; N3Q0P0 ; N3Q0P1; N3Q1P1 ; N3Q1P2 ; N3Q2P2 ; N3Q2P3 ; N3Q3P3 ; N3Q1P3; N3Q2P3;N3Q3P2.

Answer: 3414 test design 14 processing is because it is a special design based on Latin square design, this design can effectively control the test error and improve the accuracy of the test.

7.Line 173 and 174. Under this title explain the amount of N, P and K o the soil as a result; however, there is not information in material of methods about how to do this research

Answer: Soil total nitrogen content was determined by NY/T 53-1987 standard Kjeldt nitrogen determination method, soil total phosphorus content was determined by NY/T 88-1988 standard molybdenum-antimony resistance colorimetric method, and soil total potassium content was determined by NY/T 87-1988 standard acid melting flame photometer and FP6450 type flame photometer.(line147-152)

8.Line 216 and 217 both cite that fertilizer increase the yield per plant

Answer: “it can be seen that N and P fertilizers can promote the increase of yield per plant ,and the yield per plant of L. japonica gradually increases with the increase of fertilizer amount. ”instead“With the increase of nitrogen and phosphate fertilizer amount, the yield per plant of L. japonica increased gradually. ”(line220-222)

9.The authors should check the references. For example: Number 23 is not cited in the text.

Answer:The original has been amended.(line357)

10.Line 319. Delete one standard deviation

Answer: Deleted from the original text.(line322)

11.Line from 340 to 349 is information according to introduction section no discussion section.

Answer: Thank you for your valuable suggestions� We have corrected the manuscript as you required. “In this study, the yield of L. japonica increased with the increase of nitrogen and phosphorus content.”(line357-359)

12.Line 339 please put L. Japonica in cursive in the discussion and conclusion section.

Answer: The original has been amended.(line368,373,388)

13.Line 388-390. The two sentence join by a comma has not sense together.

Answer: Thank you for your valuable suggestions� We have corrected the manuscript as you required. “The results indicated that phosphorus (P) fertilizer was the primary limiting factor for enhancing the yield of L. japonica buds. Therefore, it is essential to consider the amount of phosphorus fertilizer applied to achieve high yields and a finding that aligns with the results of Zeng Huijie et al.”(line388-392)

14.Line 413: in summery…. has information that not was previously mentioned such as weather variation or texture.

Answer: Thank you for your valuable suggestions� We have corrected the manuscript as you required. “In summary, due to soil texture, and fertility, fertilization strategies should be tailored to the specific conditions of each planting region. Additionally, it is essential to ensure a balanced application of inorganic fertilizers to prevent soil compaction and imbalances in microbial flora, thereby enhancing the development of the L. japonica industry.”(line418-422)

---

## [Decision Letter · Decision Letter 4]

14 Jan 2025

PONE-D-24-06358R4Study on Nitrogen, Phosphorus and Potassium Formula Fertilization Model of Lonicera japonicaThunb in hilly and gully Loess Plateau of ChinaPLOS ONE

Dear Dr. Jiang,

Thank you for submitting your manuscript to PLOS ONE. After careful consideration, we feel that it has merit but does not fully meet PLOS ONE’s publication criteria as it currently stands. Therefore, we invite you to submit a revised version of the manuscript that addresses the points raised during the review process.

We look forward to receiving your revised manuscript.

Kind regards,

Susmita Lahiri (Ganguly)

Academic Editor

PLOS ONE

Reviewers' comments:

Reviewer's Responses to Questions

**Comments to the Author**

1. If the authors have adequately addressed your comments raised in a previous round of review and you feel that this manuscript is now acceptable for publication, you may indicate that here to bypass the “Comments to the Author” section, enter your conflict of interest statement in the “Confidential to Editor” section, and submit your "Accept" recommendation.

Reviewer #4: All comments have been addressed

Reviewer #8: All comments have been addressed

Reviewer #9: All comments have been addressed

2. Is the manuscript technically sound, and do the data support the conclusions?

Reviewer #4: Yes

Reviewer #8: No

Reviewer #9: Partly

3. Has the statistical analysis been performed appropriately and rigorously? 

Reviewer #4: Yes

Reviewer #8: Yes

Reviewer #9: Yes

4. Have the authors made all data underlying the findings in their manuscript fully available?

Reviewer #4: Yes

Reviewer #8: Yes

Reviewer #9: Yes

5. Is the manuscript presented in an intelligible fashion and written in standard English?

Reviewer #4: Yes

Reviewer #8: No

Reviewer #9: Yes

6. Review Comments to the Author

Reviewer #4: The author responded to the previous reviewers' comments. It looks like the author addressed all comments throughout the manuscript. However still there are some comments U added in the attached manuscript need to be addressed as well.

Reviewer #8: (No Response)

Reviewer #9: Dear authors, congratulations on the submitted manuscript. The suggestions made by this reviewer are scientific so that the manuscript can be better understood by future readers.

Title: I suggest removing the words "Study on", there is no need, since every scientific article is a study. Normally, words such as: "study, evaluation" do not attract the reader's attention.

Line 42. Remove the keyword "L. japonica". It is already in the title and should not be repeated.

Line 53. Because you used the word "artificial", it should be removed, since "cultivation" is not artificial, I have never seen artificial cultivation!

Line 62. Remove the word "biological".

Line 68. I suggest inserting a footer with the conversion of the local currency to the US dollar and the date of the conversion, or converting everything to dollars, since the manuscript will be read worldwide, and not regionally in China. There are several parts of the text that show the local currency, correct them all.

Line 97. Remove the word "artificial"

Lines 103-106. Insert more data from the soil analysis, such as cation exchange capacity, granulometry. Inform the methodologies, especially the extractors.

Line 140. Is the mass of flowers dry mass or fresh mass? It is not very clear in the methodology. The reader is confused.

Line 216. The correct value would be -0.81 and not 0.81.

Line 318. Replace "body" with "cell"

Conclusion. It is very long, it seems like a presentation of the results. The authors should summarize the text in short sentences that were concluded in the work. Example of lines 380-382: can be summarized as "Formulated fertilizers increased the productivity of L. japonica buds.

7. PLOS authors have the option to publish the peer review history of their article (what does this mean? ). If published, this will include your full peer review and any attached files.

**Do you want your identity to be public for this peer review?** For information about this choice, including consent withdrawal, please see our Privacy Policy .

Reviewer #4: No

Reviewer #8: No

Reviewer #9: No

---

## [Author Response · Author response to Decision Letter 5]

28 Feb 2025

Dear Editors and Reviewers:

We are very grateful for the further comments concerning our manuscript entitled “Study on Nitrogen, Phosphorus and Potassium Formula Fertilization Model of　Lonicera japonica Thunb in hilly and gully Loess Plateau of China (PONE-D-24-06358)”. These comments are all valuable and very helpful for the further revision and improvement of our article, which also provides an important guideline for our researches and scientific writing. We have carefully studied the comments and made corrections that we hope meet the standards for publication. If you have any questions regarding this manuscript, please do not hesitate to contact me. The detailed corrections in the article and the responses to the editors' and reviewers' comments are as follows:16. Title: I suggest removing the words "Study on", there is no need, since every scientific article is a study. Normally, words such as: "study, evaluation" do not attract the reader's attention.

Line 42. Remove the keyword "L. japonica". It is already in the title and should not be repeated.

Answer: I has been removed as requested , please refer to the original text for details.

17. Line 53. Because you used the word "artificial", it should be removed, since "cultivation" is not artificial, I have never seen artificial cultivation!

Answer: I has been removed as requested , please refer to the original text for details.

18. Line 62. Remove the word "biological".

Answer: I has been removed as requested , please refer to the original text for details.

19. Line 68. I suggest inserting a footer with the conversion of the local currency to the US dollar and the date of the conversion, or converting everything to dollars, since the manuscript will be read worldwide, and not regionally in China. There are several parts of the text that show the local currency, correct them all.

Answer: Exchange rate between US dollar and RMB on December 12, 2024, 1 USD=7.2674 CHY, 1 CHY=0.1376 USD

20. Line 97. Remove the word "artificial"

Answer: I has not been found "artificial" in 97.

21. Lines 103-106. Insert more data from the soil analysis, such as cation exchange capacity, granulometry. Inform the methodologies, especially the extractors.

Answer: The cation exchange capacity of the soil was measured at 8.69 cmol/kg. The granulometric composition of the soil was as follows: 4.0% for particles sized between 0.2 and 2 mm, 48.1% for particles sized between 0.02 and 0.2 mm, 34.3% for particles sized between 0.002 and 0.02 mm, and 13.6% for particles smaller than 0.002 mm.(Line119-123)

The cation exchange capacity was determined using the EDTA ammonium acetate exchange method (pH > 7.5), as outlined in Chapter 3, Section 5, Method 2 of the Soil Analysis Technical Specification. Additionally, the granulometry of the soil was assessed following the HJ 1068-2019 guidelines for the determination of soil particle size, utilizing both the pipette method and the hydrometer method.(Line176-181)

22. Line 140. Is the mass of flowers dry mass or fresh mass? It is not very clear in the methodology. The reader is confused.

Answer: The mass of flowers is the fresh and wet flower .(Line165-166)

23.Line 216. The correct value would be -0.81 and not 0.81.

Answer:Y=296.66-0.81X1+11.53X2+4.05X3-0.051X12-0.60X22-0.28X32+0.13X1*X2+0.20X1 * X3-0.029X2 * X3; where 0.81, 11.53 and 4.05 are the absolute values of the first term of the equation, respectively.

24. Line 318. Replace "body" with "cell"

Answer: I has been replaced as requested , please refer to the original text for details.

25. The revisions proposed by the reviewers in the PDF paper (PONE-D24-06358 R4 reviewer.pdf) have been implemented as required. The specific content has been modified in the original text to enhance clarity and precision, in alignment with academic standards. Each change aims to improve the overall quality of the manuscript, ensuring that it meets the expectations for publication.

---

## [Decision Letter · Decision Letter 5]

16 Apr 2025

Nitrogen, Phosphorus and Potassium Formula Fertilization Model of Lonicera japonicaThunb in hilly and gully Loess Plateau of China

PONE-D-24-06358R5

Dear Dr. Jiang,

We’re pleased to inform you that your manuscript has been judged scientifically suitable for publication and will be formally accepted for publication once it meets all outstanding technical requirements.

Kind regards,

Susmita Lahiri (Ganguly)

Academic Editor

PLOS ONE

Additional Editor Comments (optional):

Reviewers' comments:

Reviewer's Responses to Questions

**Comments to the Author**

1. If the authors have adequately addressed your comments raised in a previous round of review and you feel that this manuscript is now acceptable for publication, you may indicate that here to bypass the “Comments to the Author” section, enter your conflict of interest statement in the “Confidential to Editor” section, and submit your "Accept" recommendation.

Reviewer #4: All comments have been addressed

Reviewer #8: (No Response)

2. Is the manuscript technically sound, and do the data support the conclusions?

Reviewer #4: Yes

Reviewer #8: (No Response)

3. Has the statistical analysis been performed appropriately and rigorously? 

Reviewer #4: Yes

Reviewer #8: (No Response)

4. Have the authors made all data underlying the findings in their manuscript fully available?

Reviewer #4: Yes

Reviewer #8: (No Response)

5. Is the manuscript presented in an intelligible fashion and written in standard English?

Reviewer #4: Yes

Reviewer #8: (No Response)

6. Review Comments to the Author

Reviewer #4: The author responded to all comments. The manuscript can be accepted if there is no negative report from other reviewers

Reviewer #8: (No Response)

7. PLOS authors have the option to publish the peer review history of their article (what does this mean? ). If published, this will include your full peer review and any attached files.

**Do you want your identity to be public for this peer review?** For information about this choice, including consent withdrawal, please see our Privacy Policy .

Reviewer #4: No

Reviewer #8: No

---

## [Editor Report · Acceptance letter]

PONE-D-24-06358R5

PLOS ONE

Dear Dr. Jiang,

I'm pleased to inform you that your manuscript has been deemed suitable for publication in PLOS ONE. Congratulations! Your manuscript is now being handed over to our production team.

Kind regards,

on behalf of

Dr. Susmita Lahiri (Ganguly)

Academic Editor

PLOS ONE